# The Determinants of Liver Fibrosis in Patients with Nonalcoholic Fatty Liver Disease and Type 2 Diabetes Mellitus

**DOI:** 10.3390/biomedicines10071487

**Published:** 2022-06-23

**Authors:** Chia-Yen Dai, Tzu-Jung Fang, Wei-Wen Hung, Hui-Ju Tsai, Yi-Chun Tsai

**Affiliations:** 1Hepatobiliary Division, Department of Internal Medicine, Kaohsiung Medical University Hospital, Kaohsiung Medical University, Kaohsiung 807, Taiwan; daichiayen@gmail.com; 2School of Medicine, College of Medicine, Kaohsiung Medical University, Kaohsiung 807, Taiwan; 3Division of Geriatrics and Gerontology, Department of Internal Medicine, Kaohsiung Medical University Hospital, Kaohsiung 807, Taiwan; 910062@ms.kmuh.org.tw; 4Division of Endocrinology and Metabolism, Kaohsiung Medical University Hospital, Kaohsiung 807, Taiwan; hung4488@ms57.hinet.net; 5Department of Family Medicine, Kaohsiung Municipal Ta-Tung Hospital, Kaohsiung Medical University, Kaohsiung 801, Taiwan; 6Research Center for Environmental Medicine, Kaohsiung Medical University, Kaohsiung 807, Taiwan; 7Division of General Medicine, Kaohsiung Medical University Hospital, Kaohsiung 807, Taiwan; 8Division of Nephrology, Kaohsiung Medical University Hospital, Kaohsiung 807, Taiwan; 9Liquid Biopsy and Cohort Research Center, Kaohsiung Medical University, Kaohsiung 807, Taiwan; 10Drug Development and Value Creation Research Center, Kaohsiung Medical University, Kaohsiung 807, Taiwan

**Keywords:** liver fibrosis, type 2 diabetes mellitus, NAFLD, fibroscan

## Abstract

Liver fibrosis is a key pathophysiology process in chronic liver disease. It is still unclear whether the impact of liver fibrosis is not fully realized in type 2 diabetes mellitus (T2D) patients with nonalcoholic fatty liver disease (NAFLD), and the factors affecting nonalcoholic steatohepatitis (NASH) or liver stiffness also remain unclear. The aim of this study was to evaluate the determinants of liver fibrosis and in T2D patients with NAFLD. Liver fibrosis and steatosis were measured using transient elastography (FibroScan). Of 226 T2D patients with NAFLD, 50 with liver fibrosis had higher body mass index, serum uric acid, triglyceride and glycated hemoglobin levels and lower high density lipoprotein levels than 176 without liver fibrosis. Multivariate analysis revealed that aging, obesity, sulfonylurea usage and high levels of AST increased the risk of liver fibrosis in T2D patients with NAFLD. Our findings provide useful information to clinical physicians for earlier detection of liver fibrosis in T2D patients with NAFLD and to prevent liver fibrosis through controlling these risk factors.

## 1. Introduction

Non-alcoholic fatty liver disease (NAFLD) is currently the most common liver disease. NAFLD is defined by pathologic accumulation of fat in the liver, and it comprises a broad range of abnormalities ranging from simple fatty liver (steatosis) to non-alcoholic steatohepatitis (NASH) [1], while further progressing to liver cirrhosis and hepatocellular carcinoma (HCC) [2]. It is closely associated with metabolic disorders, such as type 2 diabetes mellitus (T2D), hypertension, hyperlipidemia, and obesity [3]. The age-adjusted relative risk of NAFLD is about 5.36-fold higher in the T2D population compared to healthy individuals [4], with 50–70% of T2D patients suffering from NAFLD [5,6]. NAFLD and T2D share some common pathophysiologic pathways, including insulin resistance, metabolic and oxidative stress, apoptosis, inflammation, fibrogenesis, genetic predisposition and environmental factors [7,8,9].

The presence of T2D is associated with NAFLD progression to advanced fibrosis, cirrhosis and HCC [10,11,12], with the prevalence of liver fibrosis being significantly higher in patients with T2D than in the general population [13]. Moreover, the prevalence of liver fibrosis among patients with both T2D and NAFLD is 17.02% [14], and the interaction among T2D, NAFLD and liver fibrosis is complicated. Hyperglycemia can induce deleterious effects on liver cells and initiate NAFLD progression from simple steatosis to NASH and fibrosis [15]; additionally, NAFLD increases the risk of liver fibrosis in T2D patients [16]. Untreated liver fibrosis will turn into irreversible cirrhosis and HCC, thereby leading to mortality [16]. Several biochemical parameters and diagnostic panels of liver fibrosis have been evaluated [17,18], and early screening and diagnosis of liver fibrosis in T2D patients with NAFLD could improve the long-term prognosis of patients [19]. However, few studies have focused on the risk factors for the occurrence of liver fibrosis in T2D patients with NAFLD. Accordingly, the aim of this study is to evaluate the determinants of liver fibrosis in T2D patients with NAFLD.

## 2. Materials and Methods

### 2.1. Study Participants

This observational study was conducted at a tertiary hospital in Southern Taiwan. We enrolled 226 T2D patients from October 2016 to April 2020. Patients with hepatitis B, hepatitis C, autoimmune hepatitis, and alcohol consumption >30 grams per day on average before enrollment of this study were excluded. All study subjects were enrolled in the T2D education program and the principles of diet therapy to T2D were delivered on an individual basis. The study protocol was approved by the Institutional Review Board of Kaohsiung Medical University Hospital (KMUHIRB-G(II)-20160021). Informed consent was obtained in written form from all patients, and all clinical investigations were conducted according to the principles expressed in the Declaration of Helsinki.

### 2.2. Sample and Clinical Data Collection

The definition of T2D included a history of T2D, the use of anti-diabetic agents, or blood glucose levels based on American Diabetes Association criteria. Demographics and clinical data were obtained from medical records and interviews with the patients at enrollment. Hypertension was clarified as a hypertensive history or the use of anti-hypertensive medications. The information of the use of anti-diabetic agents (sulfonylurea, metformin, dipeptidyl peptidase 4 (DPP-4) inhibitor, thiazolidinedione and insulin) and statins at enrollment was obtained from medical records. Body mass index (BMI) was calculated as body weight divided by body height squared. The patients were asked to fast for at least 12 h before blood sample collection for biochemistry studies including liver and kidney function, lipid profile, uric acid, and glycated hemoglobin.

### 2.3. Liver Steatosis and Stiffness Measurement

Liver steatosis and stiffness were measured using FibroScan (Echosens, Paris, France) at enrollment. FibroScan, as an ultrasound-based method of elastography, can evaluate liver steatosis and fibrosis by measuring the controlled attenuation parameter (CAP) and E med, respectively, with strong correlation of stage of liver steatosis and fibrosis examined by simultaneous liver biopsy [20]. NAFLD was defined as CAP > 238 dB/m, and liver fibrosis was defined as E med > 7.0 kilopascals (kPa) [21,22].

The FibroScan-aspartate aminotransferase (FAST) score was calculated based on the following formula [23]: FAST = (exp (–1.65 + 1.07 × ln (LSM) + 2.66 × 10^–8^ × CAP^3^–63.3 × AST^–1^))/(1 + exp (–1.65 + 1.07 × ln [LSM] + 2.66 × 10^–8^ × CAP^3^– 63.3 × AST^–1^)). The risk of NASH was divided into three groups: low-risk (FAST score ≤0.35), indeterminate risk (0.35 < FAST score < 0.67), and high-risk (FAST score ≥0.67) [23]. We defined increased risk of NASH as FAST score ≥0.35 [23]. The Fibrosis-4 (FIB) index was calculated as a previous study [24].

### 2.4. Statistical Analysis

The baseline characteristics of the patients were stratified by liver stiffness. Categorical variables were presented as percentages. Continuous variables were presented as mean ± SD or median (25th, 75th percentile) and those with skewed distribution were log-transformed to approximate normal distribution. Independent *t*-test or the Mann–Whitney U analysis was used to analyze the significance of differences in continuous variables between two groups, while Chi-square test was used to test differences in the distribution of categorical variables. Correlations among continuous variables were examined using Spearman correlation analysis. Multivariate logistic regression models were used to evaluate the association between clinical factors and liver stiffness. All the variables in Table 1 tested by univariate analysis and those variables with *p*-value less than 0.05, age, and sex were selected in multivariate analysis. Statistical analyses were conducted using SPSS version 22.0 for Windows (SPSS Inc., Chicago, IL, USA) and the graphs were made by GraphPad Prism 9.0 (GraphPad Software Inc., San Diego, CA, USA). Statistical significance was set at a two-sided *p*-value of <0.05.

## 3. Results

### 3.1. Characteristics of Entire Cohort

The comparison of clinical characteristics between groups based on the severity of liver stiffness is shown in Table 1. Of 226 T2D patients with NAFLD, 50 had liver fibrosis. Mean age was 62.1 ± 10.7 years, males comprised 51.8%, T2D duration was 10.0 ± 7.8 years, and median glycated hemoglobin was 6.9%, while 26.1% of subjects had a smoking habit and 19.9% had an alcohol consumption habit. The prevalence of hypertension, heart disease, and hyperlipidemia was 65.0%, 34.5%, and 81.0%, respectively. The median of E med was 5.3 (4.3, 6.4) kPa, and the means of CAP and FAST score were 294.5 ± 39.3 dB/m and 0.25 ± 0.20, respectively. Twenty-two percent of T2D patients with NAFLD had increased risk of NASH. Among anti-diabetic agents, T2D patients with NAFLD with liver stiffness had a higher prevalence of sulfonylurea usage and lower prevalence of thiazolidinedione usage compared with those without liver stiffness. There was no difference in the proportion of smoking, alcohol, or statin usage between the two groups. High BMI, serum uric acid, triglyceride and glycated hemoglobin levels, and lower high-density lipoprotein-cholesterol (HDL-C) levels, and AST and alanine aminotransferase (ALT) activities were found in T2D patients who had NAFLD with liver stiffness than those without liver stiffness.

### 3.2. Determinants of Liver Stiffness in T2D Patients with NAFLD

To examine the relationship between clinical parameters and E med, Spearman correlation was used. We found a positive correlation between E med level and BMI (r = 0.26, *p* < 0.001), serum uric acid (r = 0.23, *p* < 0.001), AST (r = 0.44, *p* < 0.001), ALT (r = 0.41, *p* < 0.001) and triglyceride levels (r = 0.13, *p* = 0.04) and negative correlation between E med and serum HDL-cholesterol level (r = −0.25, *p* < 0.001) (Figure 1). Furthermore, logistic regression was used to analyze the determinants of liver fibrosis in T2D patients with NAFLD. In univariate analysis, CAP, BMI, serum uric acid level, AST and ALT activities, and sulfonylurea usage were significantly and positively associated with increased risk of liver stiffness. Thiazolidinedione usage and serum HDL level were negatively correlated with an increased risk of liver fibrosis (Table 2). Adjusting for age, sex, CAP, BMI, sulfonylurea and thiazolidinedione usage, serum uric acid and HDL-C levels and AST and ALT activities in multivariate analysis, the results revealed that patients with older age (odds ratio (OR): 1.06, 95% confidence index (CI): 1.01–1.11), high BMI (OR: 1.14, 95% CI: 1.02–1.26), sulfonylurea usage (OR: 2.83, 95% CI: 1.22–6.57), and elevated AST activity (OR: 1.12, 95% CI: 1.05–1.19) had an increased risk of liver fibrosis in T2D patients with NAFLD.

We further stratified T2D patients with NAFLD by age of 65 years and BMI of 25 kg/m^2^ to investigate the effect of aging and being overweight on liver fibrosis (Table 3). The results revealed sulfonylurea usage increased the risk of liver stiffness in those patients with age ≥65 years or BMI ≥25 kg/m^2^. In T2D patients with NAFLD aged ≥65 years, a positive relationship between BMI and liver stiffness was found. Serum AST activity was correlated with liver stiffness in T2D patients with NAFLD independent of age and BMI.

### 3.3. Determinants of Risk of NASH in T2D Patients with NAFLD

In order to understand potential determinants of increased risk of NASH, T2D patients with NAFLD who had a FAST score of 0.35 or above were defined as being at risk for NASH. Adjusted logistic regression analysis revealed positive association of BMI and age with increased risk of NASH, meaning that older or obese T2D patients with NAFLD were more likely to have elevated risk of NASH (Table 4).

## 4. Discussion

This study investigated the determinants of liver fibrosis in T2D patients with NAFLD. We found that T2D patients with NAFLD who were of an older age and who had high BMI, sulfonylurea usage, and elevated serum AST activity had an increased risk of liver fibrosis. Further findings indicate a positive relationship between sulfonylurea usage and liver fibrosis in aging or overweight T2D patients with NAFLD. The correlation between BMI and liver fibrosis was only found in T2D patients with NAFLD with age ≥65 years. Independent of age and BMI, serum AST activity was correlated with liver fibrosis in T2D patients with NAFLD. In addition, aging and obesity were also associated with increased risk of NASH. Developing a more practical clinical strategy to identify the risk factors for liver fibrosis in T2D patients with NAFLD is very important. Our results provide useful information for earlier detection of liver fibrosis in T2D patients with NAFLD.

NAFLD, which is a multisystem disease that affects multiple organs, including hepatic, cardiovascular, and cerebrovascular systems, often presents in T2D patients [25]. In addition, T2D is considered a risk factor for progression to liver fibrosis, cirrhosis, and even HCC in NAFLD [26]. Lomonaco et al. indicated that moderate-to-advanced fibrosis (F2 or higher), an established risk factor for cirrhosis and overall mortality, affects at least 15% of patients with T2D [27]. In our study, the proportion of fibrosis was over 20% in T2D patients with NAFLD. In addition, we identified that age was positively related to NASH and liver fibrosis in T2D patients with NAFLD. Several studies have showed that age is a factor associated with the development and severity of liver fibrosis in NAFLD and T2D patients; however, other studies have indicated that the increased probability of liver fibrosis among T2D patients is independent of age [25,28,29,30,31,32]. Liver is an organ with age-related changes, and the complications from chronic liver disease increase with age [31,33]. Aging induces metabolic disturbances and promotes lipid accumulation in the liver [31]. With reduced physical activity in aging, insulin resistance and hyperinsulinemia also lead to obesity and the onset of liver fibrosis [31]. Thus, aging is a potential indicator of liver fibrosis in T2D patients with NAFLD.

Our findings show that high BMI was significantly associated with an increased risk of NASH and liver fibrosis in T2D patients with NAFLD. Previous studies have indicated that high BMI is an independent risk factor of liver fibrosis among T2D patients [25,30,32,34]. Accumulated studies further suggest a causal relationship between excess fat accumulation, insulin resistance, and progression to liver fibrosis [35]. Both obesity and hyperglycemia are risk factors for NAFLD, and more than 90% of obese patients with T2D have NAFLD [36]. Obesity contributes to the development of insulin resistance and increased adiposity, which promotes fat accumulation in the hepatocytes, lipotoxicity, and inflammation in the liver [37]. Hyperglycemia and toxic lipids might result in NAFLD progression from simple steatosis to NASH and fibrosis through various mechanisms, including oxidative stress, endoplasmic reticulum stress and mitochondrial disorders [15].

Fibrosis is developed in response to tissue injury and is accompanied by extracellular matrix accumulation secreted mainly by hepatic stellate cells [38]. Therefore, high BMI might increase the risk of liver fibrosis in T2D patients with NAFLD, and weight reduction remains the most effective therapy to prevent development of liver fibrosis. However, NAFLD in non-obese individuals has recently caught clinical physicians’ attention. The prevalence of lean/non-obese NAFLD varies widely, ranging from 3% to 30% globally [39]. Recent studies suggest that individuals with non-obese NAFLD exhibit substantial liver and non-liver comorbidities including advanced fibrosis, and cardiovascular- and liver-related mortality [39,40,41]. Insulin resistance, visceral obesity, greater waist circumference, male sex, hypertension and dyslipidemia are potential risk factors for NAFLD in the non-obese population [42]. Overall, this population also needs to undergo the medical screening for NAFLD.

Another interesting finding in our study is that T2D patients with NAFLD who used sulfonylurea had increased risk for liver fibrosis. Furthermore, a positive relationship between sulfonylurea usage and liver fibrosis was noted in aging or overweight T2D patients with NAFLD. Few studies have explored the relationship between sulfonylurea use and liver fibrosis progression [43,44,45,46]. Sulfonylureas bind to the specific sulfonylurea receptors of pancreatic β cells to inhibit K-ATP channels and stimulate insulin secretion in a glucose-dependent manner [47]. Sulfonylureas are common oral antidiabetic agents to increase circulating insulin levels, but they may increase weight and body fat mass, thereby having negative effects on liver pathology in NAFLD [48,49], although some reports have indicated an inconsistent relationship between sulfonylurea use and liver fibrosis progression [43,44]. A population-based cohort study reported no association between high risk of hepatic failure and sulfonylurea use in T2D patients with compensated liver cirrhosis [43]. As a result, the detailed relationship and mechanism between sulfonylurea and liver fibrosis in T2D patients with NAFLD require further exploration.

We found that serum AST activity was correlated with liver fibrosis in T2D patients with NAFLD independent of age and BMI. Serum ALT and AST activities have been widely used by clinicians to access the liver function and damage [50]. ALT and AST activities were found to be significantly higher in T2D patients with NAFLD when compared to T2D patients without NAFLD [51]. Previous studies have also indicated that AST activity was positively related to liver fibrosis in T2D patients with NAFLD [30,52,53], although several studies have found that many patients with NAFLD, fibrosis and cirrhosis might still have normal activities of AST and ALT [32,54]. Serum AST and ALT activities have the advantage of low cost and availability for detection of liver injury, establishing their measurement as important biomarkers in identifying the presence of liver fibrosis in T2D patients with NAFLD.

This study has several limitations. Firstly, the study design is cross-sectional, and the causal relationship could not be confirmed, and secondly, it was performed in a single center, and the sample size was relatively small. Thirdly, because our participants were all of Chinese ethnicity, our findings may not be generalizable to other ethnic groups. Finally, owing to few cases of moderate to severe fibrosis, the impact of determinants on liver fibrosis might be underestimated. Although liver biopsy is considered the gold standard for NASH or fibrosis, this technique is an invasive procedure and post-interventional complication might occur [55]; therefore, noninvasive screening tools such as FibroScan and clinical biochemistry data could be useful for clinical practice.

## 5. Conclusions

In conclusion, this study investigated the determinants of liver fibrosis in T2D patients with NAFLD. We found that T2D patients who had NAFLD and who were of older age and had high BMI, sulfonylurea usage, and elevated serum AST level had an increased risk of liver fibrosis. A more timely diagnosis of liver fibrosis is important and physicians should be aware of the risk factors related to liver fibrosis in T2D patients with NAFLD.

## Figures and Tables

**Figure 1 biomedicines-10-01487-f001:**
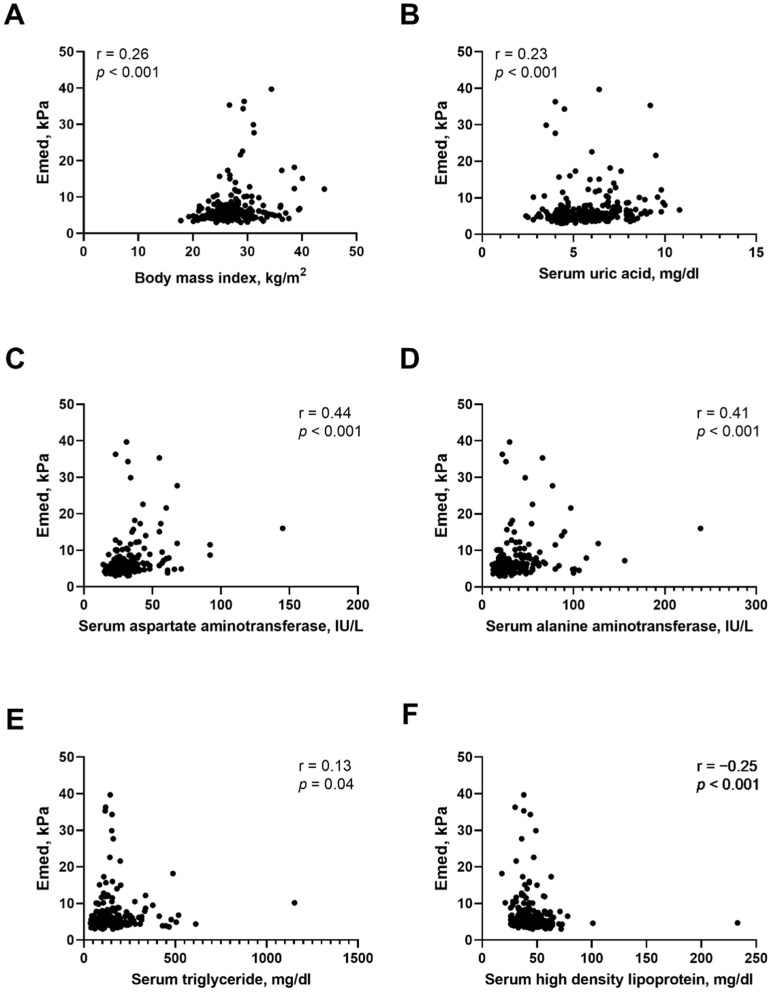
Correlation between Emed and clinical data in T2D patients with NAFLD. (**A**) body mass index. (**B**) serum uric acid level. (**C**) serum aspartate aminotransferase activity. (**D**) serum alanine aminotransferase activity. (**E**) serum triglyceride level. (**F**) serum high density lipoprotein level.

**Table 1 biomedicines-10-01487-t001:** The characteristics of study participants.

	Entire Cohort (*n* = 226)	F0-F1 (*n* = 176)	F2-F4 (*n* = 50)	*p*-Value
Age, year	62.1 ± 10.7	61.8 ± 10.5	63.1 ± 11.3	0.46
Sex (male), %	51.8	50.6	56.0	0.71
Smoke, %	26.1	25.0	30.0	0.48
Alcohol, %	19.9	18.8	24.0	0.41
Heart disease, %	34.5	35.2	32.0	0.67
Hypertension, %	65.0	63.6	70.0	0.41
Hyperlipidemia, %	81.0	82.4	76.0	0.31
T2D duration, year	10.0 ± 7.8	10.4 ± 7.6	8.6 ± 8.3	0.21
Body Mass Index, kg/m^2^	27.3 ± 4.1	26.7 ± 3.8	29.2 ± 4.5	0.001
Body Mass Index ≥25 kg/m^2^	68.6	63.6	86.0	0.003
Emed, kPa	5.3 (4.4,6.8)	4.8 (4.3,5.8)	10.4 (8.2,15.8)	<0.001
CAP, dB/m	294.5 ± 39.3	291.4 ± 38.4	305.4 ± 40.7	0.03
FIB-4	1.5 ± 0.8	1.4 ± 0.6	2.0 ± 1.0	<0.001
FAST score	0.25 ± 0.20	0.17 ± 0.12	0.50 ± 0.21	<0.001
FAST score ≥0.35, %	22.6	8.0	74.0	<0.001
Medication				
Sulfonylurea (yes vs. no)	47.3	43.8	60.0	0.04
DPP4 inhibitor (yes vs. no)	64.2	61.4	74.0	0.10
Metformin (yes vs. no)	81.0	82.4	76.0	0.31
Thiazolidinedione (yes vs.no)	38.9	42.6	26.0	0.03
Insulin (yes vs. no)	13.3	12.5	16.0	0.52
Statin (yes vs. no)	46.9	47.7	44.0	0.64
Laboratory parameters				
Creatinine, mg/dL	1.0 ± 0.5	1.0 ± 0.4	1.0 ± 0.6	0.25
Hemoglobin, g/dL	13.7 ± 1.6	13.7 ± 1.5	13.7 ± 1.8	0.76
Albumin, g/dL	4.6± 0.3	4.6 ± 0.3	4.6 ± 0.2	0.88
Uric acid, mg/dL	5.8 ± 1.6	5.7 ± 1.5	6.4 ± 1.8	0.003
AST, IU/L	30.6 ± 14.8	27.4 ± 9.8	41.8 ± 22.4	<0.001
ALT, IU/L	34.8 ± 25.9	30.7 ± 19.9	49.3 ± 37.2	<0.001
Cholesterol, mg/dL	167.7 ± 39.7	167.9 ± 39.4	167.0 ± 41.2	0.89
Triglyceride, mg/dL	127 (95,179)	123 (91,178)	148 (108,181)	0.09
HDL-C, mg/dL	44.7 ± 16.9	45.9 ± 18.2	40.3 ± 10.3	0.006
LDL-C, mg/dL	94.2 ± 33.1	93.6 ± 31.7	96.0 ± 38.1	0.69
HbA1C, %	6.9 (6.4,7.9)	6.9 (6.4,7.8)	7.6 (6.5,8.5)	0.07

Abbreviations: DPP4, Dipeptidyl peptidase 4; FAST, FibroScan-aspartate aminotransferase; AST, aspartate aminotransferase; ALT, alanine aminotransferase; HDL, high-density lipoprotein; LDL, low density-lipoprotein; CAP, Controlled Attenuation Parameter; FIB-4: Fibrosis-4; HbA1C, glycated hemoglobin.

**Table 2 biomedicines-10-01487-t002:** Logistic regression of determinants of liver fibrosis in T2D patients with NAFLD.

	Crude OR (95%Cl)	*p*-Value	Adjusted OR (95%Cl)	*p*-Value
Clinical data				
Age, year	1.01 (0.98–1.04)	0.44	1.06 (1.01–1.11)	0.02
Sex (female vs. male)	0.85 (0.46–1.57)	0.60	1.34 (0.57–3.11)	0.50
T2D duration, year	0.97 (0.92–1.02)	0.18	-	-
Body mass index, kg/m^2^	1.15 (1.06–1.24)	<0.001	1.14 (1.02–1.26)	0.02
Smoke (yes vs. no)	1.29 (0.64–2.57)	0.48	-	-
Alcohol (yes vs. no)	1.237 (0.65–2.90)	0.41	-	-
Heart disease (yes vs. no)	0.87 (0.44–1.69)	0.67	-	-
Hyperlipidemia (yes vs. no)	0.68 (0.32–1.44)	0.31	-	-
Hypertension (yes vs. no)	1.33 (0.68–2.63)	0.41	-	-
Sulfonylurea (yes vs. no)	1.93 (1.02–3.66)	0.04	2.83 (1.22–6.57)	0.02
DPP4 inhibitor (yes vs. no)	1.79 (0.89–3.61)	0.10	-	-
Metfotmin (yes vs. no)	0.68 (0.32–1.44)	0.31	-	-
Thiazolidinedione (yes vs. no)	0.47 (0.24–0.95)	0.04	0.65 (0.27–1.56)	0.34
Insulin (yes vs. no)	1.33 (0.55–3.21)	0.52	-	-
Statin (yes vs. no)	0.86 (0.46–1.62)	0.64	-	-
Laboratory data				
CAP, dB/m	1.01 (1.00–1.02)	0.03	1.01 (0.99–1.02)	0.15
Creatinine, mg/dL	1.51 (0.82–2.77)	0.18	-	-
Hemoglobin, g/dL	1.04 (0.85–1.27)	0.73	-	-
Albumin, g/dL	1.08 (0.41–2.88)	0.88	-	-
Uric acid, mg/dL	1.34 (1.10–1.64)	0.004	1.20 (0.94–1.53)	0.15
AST, IU/L	1.07 (1.04–1.10)	<0.001	1.12 (1.05–1.19)	<0.001
ALT, IU/L	1.03 (1.01–1.04)	<0.001	0.98 (0.95–1.01)	0.24
Cholesterol, mg/dL	1.00 (0.99–1.01)	0.89	-	-
Log (Triglyceride)	3.36 (0.90–12.48)	0.07	-	-
HDL-C, mg/dL	0.96 (0.93–0.99)	0.01	0.97 (0.93–1.01)	0.17
LDL-C, mg/dL	1.00 (0.99–1.01)	0.66	-	-
HbA1C, %	1.00 (0.93–1.07)	0.96	-	-

Abbreviations: OR, odds ratio; DPP4, Dipeptidyl peptidase 4; CAP, Controlled Attenuation Parameter; AST, aspartate aminotransferase; ALT, alanine aminotransferase; HDL, high-density lipoprotein; LDL, low density-lipoprotein; HbA1C, glycated hemoglobin.

**Table 3 biomedicines-10-01487-t003:** Adjusted risk factors for liver fibrosis in T2D patients with NAFLD stratified by Age and BMI.

	Age ≥65 Years		Age <65 Years		BMI ≥25 kg/m^2^		BMI <25 kg/m^2^	
	Adjusted OR (95%Cl)	*p*-Value	Adjusted OR (95%Cl)	*p*-Value	Adjusted OR (95%Cl)	*p*-Value	Adjusted OR (95%Cl)	*p*-Value
Age, year	-	-	-	-	1.02 (0.98–1.07)	0.25	1.15 (0.97–1.36)	0.19
Sex(female vs. male)	2.30 (0.56–9.40)	0.25	1.05 (0.34–3.23)	0.94	1.01 (0.42–2.43)	0.98	43.56 (0.65–2942.19)	0.08
Body mass index, kg/m^2^	1.41 (1.08–1.84)	0.01	1.08 (0.96–1.22)	0.21	-	-	-	-
Sulfonylurea (yes vs. no)	11.37 (2.06–62.88)	0.005	1.64 (0.55–4.84)	0.37	2.80 (1.14–6.84)	0.02	0.89 (0.06–14.49)	0.94
Thiazolidinedione (yes vs. no)	0.47 (0.11–1.97)	0.30	0.43 (0.13–1.45)	0.17	0.47 (0.18–1.23)	0.13	1.99 (0.13–30.60)	0.62
Uric acid, mg/dL	1.39 (0.92–2.10)	0.12	1.26 (0.88–1.80)	0.21	1.22 (0.93–1.60)	0.15	2.68 (1.01–7.11)	0.04
AST	1.32 (1.10–1.59)	0.003	1.10 (1.02–1.18)	0.02	1.11 (1.04–1.18)	0.002	1.60 (1.01–2.52)	0.04
ALT	0.96 (0.87–1.05)	0.36	0.98 (0.95–1.02)	0.28	0.98 (0.95–1.02)	0.26	0.77 (0.55–1.08)	0.13
HDL-C, mg/dL	1.01 (0.94–1.09)	0.75	0.96 (0.90–1.02)	0.16	0.98 (0.94–1.03)	0.42	0.84 (0.66–1.07)	0.15

Abbreviations: OR, odds ratio; AST, aspartate aminotransferase; ALT, alanine aminotransferase; HDL, high-density lipoprotein.

**Table 4 biomedicines-10-01487-t004:** Logistic regression of determinants of NASH risk (FAST score ≥0.35) in T2D patients with NAFLD.

	Crude OR (95%Cl)	*p*-Value	Adjusted OR (95%Cl)	*p*-Value
Clinical data				
Age, year	0.97 (0.94–0.99)	0.04	1.07 (1.02–1.13)	0.01
Sex (female vs. male)	1.20 (0.65–2.22)	0.56	2.33 (0.88–6.15)	0.09
T2D duration, year	0.96 (0.91–1.01)	0.09	-	-
Body mass index, kg/m^2^	1.20 (1.11–1.31)	<0.001	1.27 (1.13–1.44)	<0.001
Smoke (yes vs. no)	0.84 (0.41–1.74)	0.63	-	-
Alcohol (yes vs. no)	1.14 (0.53–2.45)	0.74	-	-
Heart disease (yes vs. no)	0.83 (0.431.62)	0.59	-	-
Hyperlipidemia (yes vs. no)	1.13 (0.50–2.53)	0.77	-	-
Hypertension (yes vs. no)	0.88 (0.46–1.68)	0.70	-	-
Sulfonylurea (yes vs. no)	1.64 (0.87–3.08)	0.12	-	-
DPP4 inhibitor (yes vs. no)	1.45 (0.74–2.86)	0.28	-	-
Metformin (yes vs. no)	0.53 (0.25–1.09)	0.09	-	-
Thiazolidinedione (yes vs. no)	0.35 (0.17–0.73)	0.005	0.52 (1.09–1.18)	0.22
Insulin (yes vs. no)	1.57 (0.67–3.69)	0.30	-	-
Statin (yes vs. no)	0.60 (0.32–1.14)	0.12	-	-
Laboratory data				
Creatinine, mg/dL	0.99 (0.51–1.95)	0.99	-	-
Hemoglobin, g/dL	1.20 (0.97–1.49)	0.09	-	-
Albumin, g/dL	1.08 (0.41–2.87)	0.87	-	-
Uric acid, mg/dL	1.17 (0.96–1.42)	0.12	-	-
AST, IU/L	1.29 (1.20–1.40)	<0.001	-	-
ALT, IU/L	1.10 (1.07–1.14)	<0.001	1.13 (1.09–1.18)	<0.001
Cholesterol, mg/dL	1.01 (0.99–1.01)	0.17	-	-
Log (Triglyceride)	5.94 (1.57–22.40)	0.009	3.36 (0.51–22.01)	0.20
HDL-C, mg/dL	1.00 (0.99–1.02)	0.77	-	-
LDL-C, mg/dL	1.01 (0.99–1.02)	0.19	-	-
HbA1C, %	1.01 (0.95–1.08)	0.69	-	-

Abbreviations: OR, odds ratio; DPP4, Dipeptidyl peptidase 4; AST, aspartate aminotransferase; ALT, alanine aminotransferase; HDL, high-density lipoprotein; LDL, low density-lipoprotein; HbA1C, glycated hemoglobin.

## Data Availability

The data presented in this study are available on request from the corresponding author. The data are not publicly available due to privacy.

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
