# Peer review of "The Determinants of Liver Fibrosis in Patients with Nonalcoholic Fatty Liver Disease and Type 2 Diabetes Mellitus"

_biomedicines, 2022, doi:10.3390/biomedicines10071487_

Round 1

Reviewer 1 Report

The obtained results have a limited scientific value but they are of practical value. The article is well prepared in terms of content and editorial, and can be published in the presented version.

Author Response

Thank you for your comment.

Reviewer 2 Report

The association between steatosis / non-alcoholic steatohepatitis (NAFLD / NASH) and type 2 diabetes mellitus (T2DM) is well known in the current scientific scenario. While almost all patients with T2DM have liver steatosis, the prevalence of NASH in individuals with T2DM is estimated up to 35% (vs. 5% in the general population). Similarly, the prevalence of T2DM in patients with NASH reaches 45% (vs. 8% in the general population). Both diseases are characterized by multifactorial etiopathogenesis, with genetic predisposition and environmental risk factors (lifestyle) that play a crucial role in the onset and progression of the disease.

In the present manuscript, the authors investigated the determinants of liver fibrosis in a cohort of 226 patients with T2DM. Liver fibrosis was assessed by means of transient elastography; no histologic data is reported. This is a pity, not only for liver fibrosis assessment, but especially for the possibility to provide the diagnosis of NASH. Unfortunately, as it is the manuscript completely lacks novelty; furthermore, English needs to be thoroughly revised by an English native speaker.

To improve the novelty and to provide some appealing points, I may suggest calculating the FAST score (stiffness + CAP + AST) in order to non-invasively detect patients “at-risk NASH”, that are those with higher risk of progression towards cirrhosis. As a matter of fact, these patients are considered the more appropriate target for novel therapeutics trials.

Then, authors should investigate the determinants of at-risk NASH instead of “only” liver fibrosis.

Author Response

Thank you for your valuable comment. We have investigated the determinants of at-risk NASH in our T2D patients with NAFLD, and these results were listed in New Table 4.

The FibroScan-aspartate aminotransferase (FAST) score was calculated based on the following formula: FAST = {exp (–1.65 + 1.07 × ln (LSM) + 2.66 × 10–8 × CAP3 – 63.3 × AST–1)}/{1 + exp (–1.65 + 1.07 × ln(LSM) + 2.66 × 10–8 × CAP3– 63.3 × AST–1)}. The risk of NASH was divided into three groups: low-risk (FAST score ≤ 0.35), indeterminate risk (0.35 < FAST score < 0.67), and high-risk (FAST score ≥ 0.67). We defined increased risk of NASH as FAST score ≥ 0.35. Adjusted logistic regression analysis revealed positive association of BMI and age with increased risk of NASH, meaning that older or obese T2D patients with NAFLD were more likely to have elevated risk of NASH (Table 4).

Reviewer 3 Report

NAFLD has been currently the most common chronic liver disease with a high prevalence in the world. However, the pathogenesis of NAFLD, especially NASH, is still insufficiently clear, and so is the diagnosis and therapeutic strategy. Therefore, this study is very important to evaluate the determinants of liver fibrosis in T2D patients with NAFLD.

The authors concluded that aging, obesity, sulfonylurea usage and high activity of AST increased the risk for liver fibrosis in T2D patients with NAFLD which is very important for early diagnosis of liver fibrosis, ie to prevent liver fibrosis by controlling these risk factors.

The study protocol was approved by the Institutional Review Board of 68 Kaohsiung Medical University Hospital (KMUHIRB-G(II)- 20160021). Informed consent 69 was obtained in written form from all of the patients, and all clinical investigations were 70 conducted according to the principles expressed in the Declaration of Helsinki. 

Minor comment:

Instead AST and ALT levels - put AST and ALT activity

Author Response

ANS: Thank you for your suggestion. We have revised AST and ALT “levels” as AST and ALT “activity”.

Round 2

Reviewer 2 Report

The manuscript has been improved as requested; in my opinion, it is now suitable for publication in Biomedicines.